# Contribution of Sucrose Metabolism in Phloem to Kiwifruit Bacterial Canker Resistance

**DOI:** 10.3390/plants12040918

**Published:** 2023-02-17

**Authors:** Yan Wang, Zecheng Tan, Xi Zhen, Yuanyuan Liang, Jianyou Gao, Yanhui Zhao, Shibiao Liu, Manrong Zha

**Affiliations:** 1College of Biology Resources and Environmental Sciences, Jishou University, Jishou 416000, China; 2Key Laboratory of Plant Resources Conservation and Utilization, College of Hunan Province, Jishou 416000, China

**Keywords:** kiwifruit bacterial canker, phloem, sucrose, invertase, oxidative response

## Abstract

Kiwifruit bacterial canker, caused by *Pseudomonas syringae* pv. *actinidiae* (Psa), is a catastrophic disease affecting kiwifruit worldwide. As no effective cure has been developed, planting Psa-resistant cultivars is the best way to avoid bacterial canker in kiwifruit cultivation. However, the differences in the mechanism of resistance between cultivars is poorly understood. In the present study, five local kiwifruit cultivars were used for Psa resistance evaluation and classified into different resistance categories, tolerant (T), susceptible (S), and highly susceptible (HS), based on their various symptoms of lesions on the cane. Susceptible and highly susceptible varieties had a higher sucrose concentration, and a greater decrease in sucrose content was observed after Psa inoculation in phloem than in tolerant varieties. Three invertase activities and their corresponding gene expressions were detected in the phloem with lesions and showed the same trends as the variations in sucrose concentration. Meanwhile, after Psa inoculation, enzyme activities involved in antioxidant defense responses, such as PAL, POD, and CAT, were also altered in the phloem of the lesion position. With no differences among cultivars, PAL and POD activities in phloem first increased and then decreased after Psa inoculation. However, great differences in CAT activities were observed between T and S/HS categories. Our results demonstrate that sucrose content was negatively correlated with the disease resistance of different cultivars and that the increase in immune response enzymes is likely caused by increased sucrose metabolism in the phloem.

## 1. Introduction

Kiwifruit (*Actinidia* spp.) gradually became an important economic crop worldwide due to its unique flavor and nutritional value. Kiwifruit bacterial canker (KBC), caused by *Pseudomonas syringae* pv. *actinidiae* (Psa), is considered the most devastating disease to kiwifruit production and has already become an economic threat to the kiwifruit industry worldwide [1,2]. The disease’s symptoms occur on various organs of kiwifruit plants, e.g., cane, trunk, leaves, wilting vines, and flowers [3]. Among them, a milky or reddish bacterial ooze on the cane and trunk results in vine-wide death, which leads to the most economic losses [4]. Psa was first isolated in Japan in 1984 and has since spread worldwide [5,6,7,8]. At least five types of Psa populations, including 1, 2, 3, 5, and 6, have been defined based on their virulence, toxin production, and host range [9]. Among them, three biovars were deemed to result in severe damage to the global kiwifruit industry [10,11].

Most globally cultivated kiwifruit cultivars are natural hosts of Psa [2]. Many cultivars of *Actinidia chinensis*, e.g., ‘Hongyang’ and Jintao’, are considered more susceptible to biovar 3 (Psa3) than *A. deliciosa*, including ‘Miliang No.1’ and ‘Xuxiang’ [12,13]. There is a significant difference in tolerance to kiwifruit canker between species and cultivars [14,15]. There are still no cures for KBC once the infection by Psa has begun. Streptomycin and copper were demonstrated to be effective against KBC [16], but the utility was limited [17,18]. Hence, planting cultivars with Psa resistance should be the best way to avoid KBC. 

Pathogens attack plants by suppressing host immunity responses and hijacking nutrients (mainly sugar) from the host for their growth [19]. In response, plants will develop ways to limit nutrient supply and initiate defensive activities. Sucrose (Suc) is the main form of assimilates transported in the phloem [20]; it is engaged in plant defense by activating plant immune responses against pathogens [21]. Once sucrose is transported to a particular organ, it breaks down into other sugars, such as glucose, fructose, and trehalose 6-phosphate, which are involved in responses to several stresses and act as signal molecules [22,23].

Invertase (INV), the key enzyme that catalyzes the cleavage reaction of sucrose, appears to have a crucial role in response to pathogenic attacks [21,24]. According to their optimum PH and subcellular locations, INVs have been categorized into three groups: cell wall invertase (CWINV), cytoplasmic invertase (NINV), and vacuolar invertase (vAINV) [25]. CWINV activity and plant response against pathogens during plant infection have been widely discussed [21]. in the caused disease by Erwinia carotovora in carrots, CWINV showed a very rapid increase after 1 h and subsequently with an accumulation of PAL [26]. Manipulating the expression of CWINV genes resulted in a sugar content change and defense induction after bacteria infection in different species [27,28,29]. NINVs were also proven to be a part of the antioxidant system involved in the homeostasis of cellular reactive oxygen species [30]. Increased activity of NINV was observed after inoculation by the pathogen in tobacco and *A. thalianad* [31,32]. However, vAINV modulation is poorly understood and is not involved in the plant defense response [31].

The diverse varieties of kiwifruit show different resistances to KBC. In this study, our objective was to explore whether sucrose metabolism, especially invertase activity, contributes to the different resistance of KBC among cultivars.

## 2. Results

### 2.1. Comparison of the Responses of Detached Branches of Five Kiwifruit Cultivars to PSA Infection

The cultivar resistance evaluation was based on the previous study by using canes in vitro in this study. One-year-old cane pieces of five common kiwifruit cultivars were collected and used for the canker resistance identification through inoculation with Psa suspension in vitro. After removing the bark, lesions on canes of different cultivars were observed (Figure 1). There were only small lesions around the wounds of ML (Miliang No.1) and CY (Cuiyu), and the lesions were more like dehydrated hypersensitive necrotic tissues. Lesions on the canes of BM (Beimu) were also bound around the wound with minor but general browning. Compared to the above cultivars, the XJH (Xiangjihong) and HY (Hongyang) cultivars had distinctly extended lesions along the canes, showing severe browning. The lesions spread throughout the entire cane in the HY cultivar.

Based on the length of the lesions formed after the inoculation of the cane pieces, we evaluated the susceptibility of these five kiwifruit cultivars to Psa canker (Table 1). The entire phloem of the HY cane was covered with brown lesions. The symptoms of XJH were not severe as HY, but the lesions’ length still accounted for 71.70 % (21.51 ± 2.49 cm) of the whole cane. We defined these two varieties as highly susceptible (HS) varieties. The lesion length on BM was about 7.68 ± 1.43 cm, accounting for 24.77 % of the length of the entire cane, and was thus defined as a susceptible (S) variety. However, ML and CY showed significant tolerance to Psa, and the lesion lengths of these canes were 1.72 ± 0.22 cm and 3.17 ± 0.41 cm, respectively. In addition, no infection was found in the xylem, so they were defined as tolerant (T) varieties.

### 2.2. Effect of Psa Infection on Contents of Sucrose in the Phloem of Kiwifruit Branches

Sucrose content in the phloem before and 30 days after Psa infection was determined (Figure 2). Before inoculation, the sucrose concentration in the phloem of ML and CY cultivars was lower than in BM, XJH, and HY, suggesting that Psa susceptibility positively correlates with the sucrose content in the phloem. Thirty days after Psa inoculation, the sucrose content in the phloem decreased significantly in HY, XJH, and BM cultivars, which were 7.12, 5.08, and 5.89 lower than before inoculation, respectively. However, ML and CY decreased only by 3.63 and 1.09, and the difference was insignificant. During the 30 d innoculation of Psa, sucrose consumption in susceptible varieties was higher than in tolerant varieties.

### 2.3. Effect of Psa Infection on Invertase Activity in the Phloem of Kiwifruit Branches

Invertase irreversibly hydrolyzes sucrose into glucose and fructose and has been reported to be affected during plant–pathogen interactions. The activity of three sucrose invertases (INV) in the phloem was determined 30 days after inoculation (Figure 3). After Psa inoculation, CWINV’s activity in the phloem of resistant and susceptible cultivars was significantly higher than that of canes without Psa inoculation. However, the CWINV’s activities in the phloem of highly susceptible cultivars did not show a significant difference before and after inoculation. The increase in CWINV activity after inoculation in phloem displays positive correlation with the disease tolerance of varieties. The vAINV enzyme’s activity shows different variations during infection, which are increased in ML1 and XJH and reduced in BM and HY. The NINV enzyme’s activities were significantly decreased in susceptible and highly susceptible varieties, whereas there was no significant difference in ML and CY. These results suggest that decreased NINV activity may be one of the causes of susceptibility to PSA infection.

### 2.4. Response of Sucrose Invertase Biosynthesis Genes Expression to Psa Infection

We further determined the expression of three sucrose invertase biosynthesis genes in the phloem of canes. Compared to the canes without Psa inoculation, the expression of *AcCWINV* in the phloem of each variety after PSA infection was significantly upregulated, among which ML, CY, and BM showed a greater induction of *AcCWINV*. The *AcNINV* expression level did not show significant difference in resistant and susceptible cultivars but were downregulated in BM and HY after Psa inoculation. The expression of vAINV was only upregulated in ‘ML 1’ induced by PSA infection, and no significant difference was observed in other varieties. These results were consistent with the INV enzyme activities of change regularity, indicating that the expressions of the AcCWINV gene and AcNINV gene in the phloem of detached branches of susceptible and highly susceptible varieties were regulated after PSA infection. In contrast, the expression of AcvAINV was not affected.

### 2.5. Defense Enzyme Activities in Detached Branches of Five Kiwifruit Cultivars Infected with PSA

Plant disease resistance is highly related to the activity of antioxidant defense responses and lignin synthesis. Due to the different levels of sucrose consumption in the phloem of each cultivar after Psa infection, enzyme activities were also detected in the phloem (Figure 3). During the 30 days of infection, PAL and POD activities in phloem rise at the beginning and then decline over time. Still, the extents of these changes were different between tolerant, susceptible, and highly susceptible varieties. In general, the PAL and POD activities of ML (T) and CY (T) were higher than those of other varieties. In contrast, the highly susceptible varieties have the lowest POD activities among all five cultivars.

Different from PAL and POD, CAT activity in the phloem varies with varieties for 30 days of infection. In susceptible and highly susceptible varieties, CAT activity gradually increased over time. However, in resistant varieties, CAT activity almost tripled after ten days of inoculation and then reduced to a similar level to the other varieties after the next 20 days. These results suggest that the quick response of CAT activity to Psa inoculation might be the reason for better canker resistance.

## 3. Materials and Methods

### 3.1. Plant Materials and Growth Conditions

Five kiwifruit cultivars, ‘Miliang No.1’, ‘Cuiyu’, ‘Beimu’, ‘Xiangjihong’, and ‘Hongyang,’ were selected from the experimental base for kiwifruit variety breeding and seedling production in Jishou, Hunan province, China, in January 2021. One-year-old shoots in the dormant period after their leave fell were collected from the five varieties. Twenty shoots were chosen from every variety, approximately 30 cm in diameter. Shoot samples were taken from the isolated branches 0, 10, 20, and 30 days after inoculation with PSA. Samples of 0.1 g of phloem from these isolated branches were weighed. The samples were immediately placed in liquid nitrogen and stored at −80 °C for sugar content and enzyme activity detection, RNA extraction, and further analysis. Each sample contained three shoots from 3 different vines as biological replicates.

### 3.2. PSA Activation and Inoculation

The pathogen was a highly pathogenic PSA strain type 3 isolated from the Guangxi Zhuang Autonomous Region of China, provided by the Wang Farming Team (Guangxi Institute of Botany, Chinese Academy of Sciences). Monoclonal strain PSA3 was selected from preserved glycerol tubes and cultured for 48 h on LB solid medium at 25 °C. After that, a single colony was established and grown in 100 mL LB liquid medium for about 48 h at 25 °C under shaking conditions. The final concentration was adjusted to 1 × 10^9^ CFU/mL with sterile water. Before inoculation, shoots needed to be washed with running water. Their surfaces were sterilized with freshly prepared 75% ethanol after drying, washed twice with sterile distilled water, and air-dried before inoculation. Each shoot was punched with a hole-punch of 0.5 cm in diameter deeply into the vascular cambium in the middle of the branch. Ten microlitres of a bacterial suspension solution was added to the wound. Each branch was punched the same way, but with sterile distilled water was added as a control. The inoculated branches were left for 15 min until the wound completely absorbed the bacterial solution without any visible solution remaining on the surface. Subsequently, all inoculated canes were put on a seedling tray draining board and covered with two layers of sterile absorbent paper in advance. The lower tray was filled with sterile water to the bottom of the draining board. Finally, two layers of sterile absorbent paper were placed over the cane pieces. Control cane pieces were treated in another sterile tray in the same way. The inoculated and control cane pieces were then transferred to a sterile chamber with 90% humidity and under a temperature of 10 °C. Incubators were checked every three days to ensure that the covering paper remained moist.

### 3.3. Assessment of Different Cultivars’ Disease Tolerance

After 30 days of inoculation and culture, the cortex of the branches was cut off with a sterile art knife to observe and determine the degree of infection from the pathogenic bacteria. According to the degree of branch browning, namely, the length of branch browning, the tolerance of the kiwifruit variety to canker was classified into three grades, as follows. Tolerance (T): the degree of browning near the wound is slight, which is consistent with the sterile water control treatment, and bacteria did not infect the xylem. Susceptible (S): 1/4~1/2 or 7.5~15.0 cm browning on the branches, part of the xylem near the wound also browning. Highly susceptible (HS): half or more than 15.0~30.0 cm of the branches browning, and most of the xylem evidently browned. The average numbers across five branches were used in the evaluation.

### 3.4. Detection of Sucrose in the Phloem

We used a detection kit purchased from Beijing Solarbio Life Sciences to detect plant sucrose content. From five in vitro kiwifruit cultivars, 0.1 g of branch phloem was extracted. The phloem samples were ground at room temperature, adding 1 mL of kit extract. Following that, an 80 °C water bath extraction was conducted for 10 min. Samples were centrifuged at 4000× *g* and 25 °C for 10 min. Carbon powder was added to the supernatant and decolorization was observed for 30 min; then, 1 mL extract was added and the mixture was centrifuged again at 25 °C and 4000× *g* for 10 min until the final supernatant was obtained. The optical density of the supernatant represents the sucrose content, which was colorimetrically measured at 480 nm by UV–Vis spectrophotometer.

### 3.5. Enzyme Activities Analysis

The method to extract POD, PAL, CAT, CWINV, vAINV, and NINV was similar to using enzyme reagent kits. Extracts of 1 mL of defense enzyme (POD, PAL, CAT) and sucrose invertase (CWINV, vAINV, NINV) were added to the 0.1 g of phloem from the detached branches. The samples were ground in liquid nitrogen, homogenized in an ice bath, and low-temperature centrifuged at 4 °C for 10 min (8000 r/min). The supernatant liquid was collected, and the determination of related enzyme activity was conducted according to enzyme kit instructions. The enzyme determined the final reaction products at different λ using a UV–Vis spectrophotometer (unit: Ug^−1^ (FW) min^−1^).

### 3.6. RNA Extraction and Quantitative RT-PCR

Total RNA was extracted from the roots and leaves of potato plants by using an E.Z.N.A.^®^ Total RNA Kit I (Omega Bio-Tek, Norcross, GA, USA). As previously described, a 5 μg aliquot of total RNA was used for library preparation [34].

Total RNA samples were treated with RQ1 DNase (Promega, Madison, WI, USA) for 30 min to remove genomic DNA and then converted into cDNA using iScript™ Reverse Transcription Supermix (Bio-Rad, Hercules, CA, USA). Quantitative real-time PCR (qRT-PCR) was conducted in a CFX Connect Real-Time System with iTaq Universal SYBR Green Supermix (Bio-Rad). The thermal conditions were 95 °C for 3 min, then 40 cycles of 95 °C for 15 s and 60 °C for 60 s, followed by melt curve analysis to verify the specificity of amplification. The ΔΔCt method was used as an internal control to analyze expression levels with the potato actin gene (XM_006350963).

### 3.7. Statistical Calculations

Results were analyzed using SPSS software v16.0 for Windows. Data from each sampling event were analyzed separately. Significant differences between treatments were determined by analysis of variance (ANOVA), followed by Fisher’s LSD test (*p* < 0.05). Tukey–Kramer HSD tests were used to compare the difference of the lesion lengths among genotypes (*p* < 0.05).

## 4. Discussion

In this study, we aimed to identify the disease mechanism of bacterial canker in kiwifruit. According to the symptoms of the detached branches inoculated with Psa, five local kiwifruit varieties were classified as tolerant (ML 1, CY), susceptible (BM), and highly susceptible (XJH, HY) varieties. The change in the sucrose contents, invertase enzymes, and resistance enzymes in the phloem of five kiwifruit cultivars after inoculating Psa were detected, and their relationships with BKC resistance were studied.

Nutrients acquired from the host are essential for a pathogen to cause disease. After infection, pathogens obtain carbon resources from host plants for their growth and reproduction. Pathogens will redistribute the sugar of plants according to their own needs, forcing plans to change their sugar content and triggering their defense response [21]. Previous studies have shown that increasing sucrose content contributes to plant disease tolerance in some species, such as in corn and rice [35,36]. Interestingly, the sucrose content in the phloem of susceptible and highly susceptible cultivars was higher than that of resistant varieties in our study (Figure 2). Even though our results showed a negative correlation between the tolerance of cultivars to Psa and sucrose contents in the phloem, it is not meant to be contrary to previous studies. For the same cultivar, sucrose content in the phloem varied with seasons, and a higher sucrose content always comes with higher KBC resistance (data not shown). Between cultivars, higher sucrose contents in phloem represent more carbon resources for the pathogen, which might be conducive to the proliferation of Psa. At the same time, the consumption of sucrose in susceptible varieties was much higher than that of the tolerant varieties after Psa inoculation (Figure 2).

Pathogenic infection increases the invertase enzyme activity in host plants [37], which helps to regulate sugar distribution and provide essential sugars for the reproduction of pathogens. Of note, glucose converted from sucrose under the action of CWINV also functions as a signaling molecule that can trigger the defense responses of plants, such as gene expression of pathogenesis-related proteins, cell wall reinforcements, apoptosis, and so on [21]. Numerous studies claim that enhanced CWINV activity contributes to the tolerance of bacterial diseases in some varieties [27,29,38]. Overexpression of the CWINV gene in rice could strengthen the resistance of rice to *Xanthomonas oryzae* PV. *Oryzae* and *Magnaporthe oryzae* rice [28]. In tobacco, resistance to phytophthora nicotianae was found to be decreased in CWINV-RNAi plants when compared to wild-type plants [31]. In our study, the CWINV enzyme activity and gene expression were both increased obviously after Psa infection. They were shown to be stronger in tolerant cultivars than that in susceptible and highly susceptible cultivars (Figure 3 and Figure 4). These results indicate that the tolerance of the kiwifruit branches was tightly associated with CWINV activity and gene expression.

At the same time, cell wall lignification in the infection area could inhibit the penetration of pathogenic bacteria through the cell wall, which helps prevent its further spread [39]. PAL is the key rate-limiting enzyme of the phenylathe lanine metabolism pathway, providing precursors for lignin synthesis. PAL mRNA was shown to be induced by CWINV expression in carrot [40], which resulted in the enhanced activity of PAL after inoculation in our results (Figure 5). Meanwhile, POD catalyzes the last step of lignin synthesis, and a great enhancement response to Psa infection was detected in our results. Hence, by the synthesis of phenolic compounds and, later, lignin, plants produce chemical and physical barriers against Psa. This suggests that Psa can alter the distribution of sugar in plants through infection activities, such as reducing the sucrose content and the defensive response of plants to its reproduction.

NINVs were considered to regulate plant growth and development; they also contributes to the plant metabolic signal processes response to stress [41]. In Arabidopsis, NINV-deficient mutants showed a phenotype of oxidative stress, suggesting that NINVs are part of the antioxidant system involved in the homeostasis of cellular reactive oxygen species [30]. Numerous reactive oxygen species accumulated in the host plant under pathogenic attack, causing the catastrophic death of cells [42]. CAT plays a significant role in removing reactive oxygen species that damage plant cells [43], which is necessary to maintain normal metabolism. In our result, the reduction in the gene expression and enzyme activity of NINVs was only detected in susceptible and highly susceptible cultivars after Psa inoculation (Figure 3c and Figure 4c), suggesting that NINVs might be one of the reasons for the Psa resistance difference between cultivars. When Psa invades a susceptible kiwifruit plant, it will suppress the activity of NINV and subsequently impair the antioxidation reaction of plants, providing an excellent atmosphere for Psa reproduction.

## 5. Conclusion and Indications for the Future

Understanding plan–-pathogen interactions is essential for achieving sustainable crop production. Plant–pathogen interactions are significantly influenced by the amount of sucrose content in the host plants. Upon infection by Psa in kiwifruit, the plant activates various defense systems to prevent or slow down the pathogen’s proliferation. Unsurprisingly, sucrose content is an important factor affecting the resistance of kiwifruit plants to KBC. It would be interesting to explore the role of sucrose homeostasis in the intercellular space response to Psa via a study of sugar transporter proteins, e.g., SWEET (sugar will eventually be exported transporters), STP (sugar transport protein), and SUT (Sugar transporters). Another factor that needs to be considered in the future is the relationship between sugar content and other disease-resistance factors. For example, it is known that phytohormones, e.g., salicylic acid (SA), jasmonic acid (JA), and ethylene (ET), may be involved in plant resistance. It would be interesting to explore whether sucrose signalling leads to variations in the content of immunity-related phytohormones.

## Figures and Tables

**Figure 1 plants-12-00918-f001:**
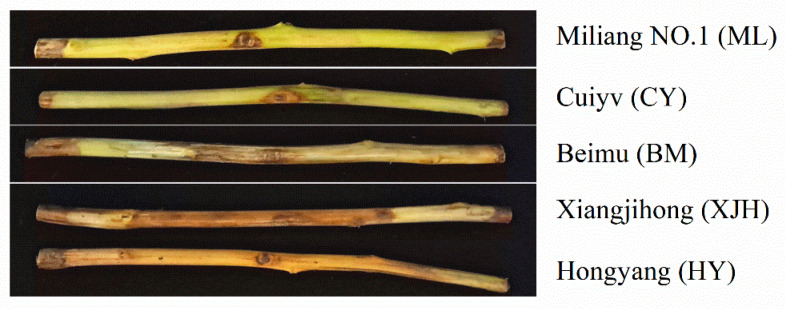
Symptoms of lesions on canes of different cultivars after 30 days inoculation with *Pseudomonas syringae* pv. *actinidiae* (Psa).

**Figure 2 plants-12-00918-f002:**
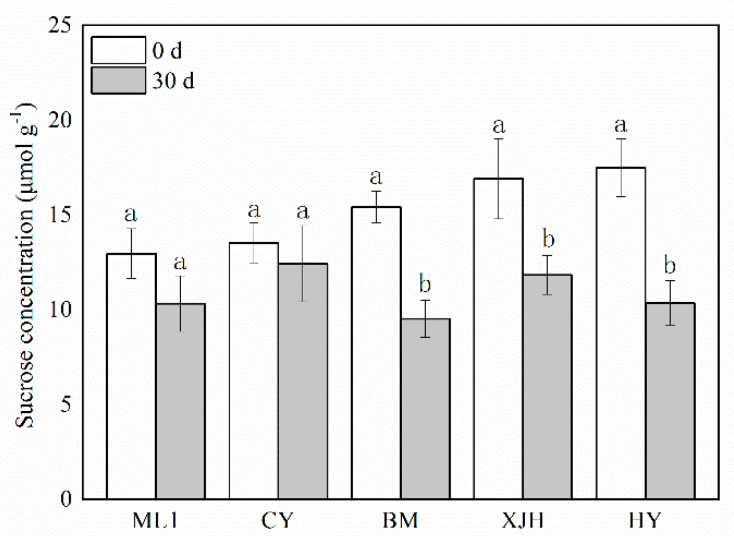
The content of sucrose in the phloem of five kiwifruit cultivars canes after 30 days of inoculation of *Pseudomonas syringae* pv. *actinidiae* (Psa). The data represent means from three biological replicates. Error bars = SD. Columns with different letters are significantly different at *p* < 0.05.

**Figure 3 plants-12-00918-f003:**
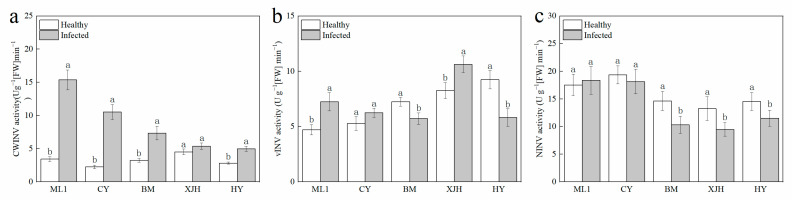
Activities of CWINV (**a**), vINV (**b**), NINV (**c**) invertase in the phloem of five kiwifruit cultivars canes after 30 days inoculation of *Pseudomonas syringae* pv. *actinidiae* (Psa). The data represent means from three biological replicates. Error bars = SD. Columns with different letters are significantly different at *p* < 0.05.

**Figure 4 plants-12-00918-f004:**
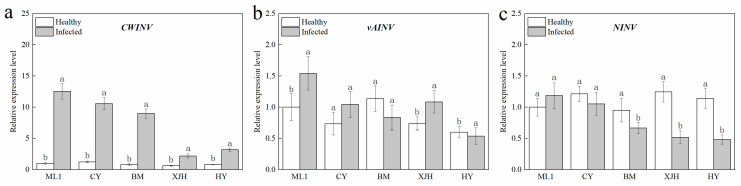
The relative expression level of *CWINV* (**a**), *vAINV* (**b**), and *NINV* (**c**) in the phloem of five kiwifruit cultivars canes after 30 days inoculation of *Pseudomonas syringae* pv. *actinidiae* (Psa). The data represent means from three biological replicates. Error bars = SD. Columns with different letters are significantly different at *p* < 0.05.

**Figure 5 plants-12-00918-f005:**
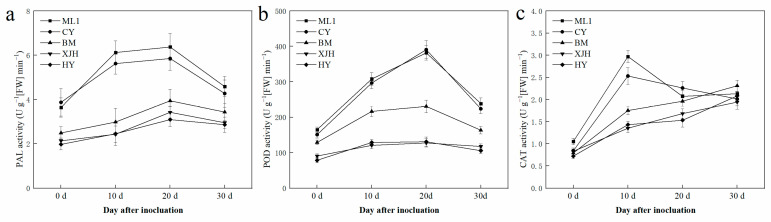
PAL (**a**), POD (**b**), and CAT (**c**) activity in the phloem of five kiwifruit cultivars canes after 30 days inoculation of *Pseudomonas syringae* pv. *actinidiae* (Psa). The data represent means from three biological replicates. Error bars = SD. Columns with different letters are significantly different at *p* < 0.05.

**Table 1 plants-12-00918-t001:** Response of five kiwifruit cultivars to in vitro inoculation with Pseudomonas syringae pv. actinidiae (Psa).

Cultivars	PSA Invasion into Xylem	ALLT (Mean ± SD)	Response to Psa
Miliang NO.1	None	1.72 ± 0.22 a	T
Cuiyv	None	3.17 ± 0.41 b	T
Beimu	Partly	7.68 ± 1.43 c	S
Xiangjihong	Partly	21.51 ± 4.69 d	HS
hongyang	Whole	30.00 ± 0.00 e	HS

Note: Mean ± SD in the table shows the means and their standard deviation. Abbreviations: HS, highly susceptible; S, susceptible; T, tolerant. The average lesion lengths of the three most seriously affected cane pieces (ALLT) were calculated as in Wang et al. [33]. The mean ALLT values followed by different letters are significantly different at *p* < 0.05 according to Tukey–Kramer HSD test.

## Data Availability

The data presented in this article are available on request from the corresponding authors.

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
