# Peer review of "Contribution of Sucrose Metabolism in Phloem to Kiwifruit Bacterial Canker Resistance"

_plants, 2023, doi:10.3390/plants12040918_

Round 1
Reviewer 1 Report
The manuscript is very important in field
But need improved language editing
Abstract not clear
In part result you used ANOVA or t test it not clear
You say in part stastical analysis used t test, while in figure appear symbols how get it, please repeat all stastical again
In part discussion where the mechanism which ocuure
Where the conclusion
Author Response
Thank you for your comments.
About the English writing of the manuscript, we asked for a native English speaker to revise the paper this time.
The abstract part was revised to make it easier to understand.
We use ANOVA to repeat all statistical again, and corrections in the manuscript.
We rewrite some sentences in the discussion part (line 269-270, 285-286, 292-297, 307-310).
We add the conclusion.

Reviewer 2 Report
no
Author Response
Thank you!
Reviewer 3 Report
The overall impression of the contribution of the current study is reasonable. However, the Authors may consider doing necessary amendments to the manuscript for better comprehensibility of the study.
1. Information and role of sugar transporters, including STP (sugar transport protein) is missing.
2. Any role of CWIN-mediated salicylic acid or jasmonic acid signaling and programmed cell death pathways?
3. Any role of a sugar-based regulation of plasmodesmatal development or aperture?
4. Please provide diagrams denoting the procedure and the mechanism.
Author Response
Thank you for your great comments.
In this work, we make a point that sucrose may contribute to the kiwifruit bacterial canker resistance. Sugar transporters like SWEETs, SUTs, and STPs are important in regulating intercellular sucrose contents. It is valuable but not necessary in this story. Actually, that’s one of the important parts we will focus on next.
Of course, Yse. We didn’t have any results about phytohormones or PCD, so we didnt talk about it .We can only discuss the function of CWIN based on our data.
Enhancing CWIN and STP activity could generate more Glc to block the plasmodesmata aperture and hence the cell-to-cell spread of virus. But for pathogenic bacteria, mainly reside in the intercellular space.
What we do in this study can only prove that sucrose takes part in Kiwifruit bacterial canker resistance. The discussion of mechanisms is not deep, and the evidence is not strong enough to give a diagram denoting the mechanism.

Round 2
Reviewer 1 Report
The author reply all comments
So I agree for published
Reviewer 3 Report
The authors have addressed my queries.